# Adolescents’ Exposure to Online Risks: Gender Disparities and Vulnerabilities Related to Online Behaviors

**DOI:** 10.3390/ijerph18115786

**Published:** 2021-05-27

**Authors:** Elena Savoia, Nigel Walsh Harriman, Max Su, Tyler Cote, Neil Shortland

**Affiliations:** 1Community Safety Branch of the Emergency Preparedness, Research, Evaluation, and Practice Program, Division of Policy Translation and Leadership Development, Harvard T.H. Chan School of Public Health, Boston, MA 02115, USA; nharriman@hsph.harvard.edu (N.W.H.); masu@hsph.harvard.edu (M.S.); 2Department of Biostatistics, Harvard T.H. Chan School of Public Health, Boston, MA 02115, USA; 3Operation250, Lowell, MA 01854, USA; tcote@operation250.org; 4Center for Terrorism and Security Studies, University of Massachusetts Lowell, Lowell, MA 01854, USA; Neil_Shortland@uml.edu

**Keywords:** online safety, online risk, online behaviors, gender disparities, high school students, social media

## Abstract

In the last decade, readily available electronic devices have created unprecedented opportunities for teens to access a wide variety of information and media–both positive and negative–on the internet. Despite the increasing number of initiatives taking place worldwide intended to assess and mitigate the online risks encountered by children and adolescents, there is still a need for a better understanding of how adolescents use the internet and their susceptibility to exposure to risks in the online space. We conducted a cross-sectional online survey of a convenience sample of 733 8th and 9th grade students in Utah. The survey contained eight questions regarding students’ exposure to three types of online risk scenarios: content risk, contact risk, and criminal risk. Independent variables included students’ online behaviors, use of social media and private messaging apps, and adult supervision of online activities. Logistic and negative binomial regression models indicated that gender, social media use, and chatting with strangers were associated with exposure to multiple risky online scenarios. Our results provide critical information to educators involved in the development of initiatives focusing on the reduction of youth online risk by identifying correlates of risky online events, allowing them to tailor their initiatives to meet the needs of potentially vulnerable populations.

## 1. Introduction

In the last decade, readily available electronic devices such as computers, tablets, and smartphones have created unprecedented opportunities for teens to communicate, connect, share, and access a wide variety of information and media on the internet [1,2]. A 2018 survey reported that nearly all U.S. teens (95%) have access to a smartphone and 45% are “almost constantly” on the internet. Information and communication technologies are playing a key role in fulfilling adolescents’ emotional and communication needs [3], and the instrumental and social functions of such technologies are having a critical impact on teens’ interactions with parents, peers, and social groups [4]. Digital communication can both strengthen and strain teen-parent relationships. Mobile phones provide a permanent channel of communication between parents and teenagers, thereby offering opportunities to intensify parental supervision and control [5,6,7]. However, the same technology is also facilitating the process by which teens connect with peers and with individuals in an independent manner free from parental surveillance.

Despite the increasing number of initiatives taking place worldwide intended to assess and mitigate the online risks encountered by children and adolescents, there is still a need for a better understanding of how adolescents use the internet and what consequences they may face [2,8]. For example, it is important to expand the existing research into the role adults play in being a protective factor to the risks and threats presented to youth online. Existing research focusing on the mediation rules and strategies of parents has found that restrictions of online peer-to-peer interaction are effective, though the most common of strategies done by parents have been found to show no significant impact [9]. Alas, the current understanding of how young people experience communication technologies, how they perceive opportunities for risks over the internet, and how social media shapes their interactions is developing but admittedly incomplete.

### 1.1. Adolescents’ Exposure to Risky Online Situations

Risky online situations that adolescents may encounter have been classified into three groups: (1) “content risk,” referring to user exposure to possible harmful contents, (2) “contact risk,” referring to user activities and communications with known or unknown individuals that carry a potential threat and (3) “commercial risk,” referring to situations in which commercial organizations attempt to exploit the user [10]. In reviews of what risks would bother fellow adolescents of their age, youth identified pornographic and violent content as being their two highest concerns (classified as content risk) [11]. As a highly interactive platform, social media facilitates more opportunities for adolescents to engage in risky behaviors and be exposed to risky online content. When online, adolescents devote much of their free time to social media, using social media platforms for an average of nearly 3 h each day [12,13]. Understanding the relationship between social media use and risky behaviors during adolescence is crucial given the increased propensity for risk-taking that is unique to this developmental period [14,15]. Several studies and reviews of existing literature provide evidence illustrating how the use of social media impacts–both positively and negatively–essential aspects of adolescents’ mental health and psychosocial development, including self-esteem, social connectedness, peer communications [16,17,18], and how social media may pose risks of cyber victimization and lurking [19,20,21,22]. A recent meta-analysis showed that higher levels of social media use among adolescents are associated with more frequent engagement in drug use, risky sexual acts, and violent behaviors with a small-to-medium magnitude of effect [23]. Specific examples of negative situations experienced by adolescents in the online space include: being contacted by a stranger, having either seen or received media that made them feel uncomfortable, having felt under pressure to send photos or other information about themselves, and having accidentally spent money online that they did not mean to spend [24,25]. As part of a three-year research effort to examine childrens’ safe use of the internet and digital technologies in Europe, researchers found that the most common of risks experienced by youth was giving out personal information, encountering violent and hateful contact being third most prevalent, and being bullied the fourth most common [26]. While all these situations have been widely documented, little is known about which individual-level online behaviors and which social media and communication platforms are most likely associated with negative and risky situations.

A range of work has sought to understand the risk factors associated with encountering harmful material online. For example, Suler identifies the role that the experience of being “online” plays in creating a risk-shift in individuals (referred to as the online disinhibition effect) [27]. Other field research has explored the base-level predictors that someone (especially a young person) will be on a social networking site and thus exposed to risk. This work equally identifies the avenues to practically manage online risks to children–via the industry providers, parents, and the children themselves. Wider meta-analyses have sought to identify the prevalence of risk; for example, noting that one in five youth had experienced unwanted exposure to sexually explicit material while online and one in nine youth experience online sexual solicitation [28]. Taken together, such work emphasizes the risk of being online in general, the nature of “being online” which increases the likelihood of risky behavior, and even the immensely high prevalence of risky experiences that are encountered by those (especially young children) who go online. While crucially important, the issue with such work is that it often draws a homogenous, and indeed static, perception of internet users, meaning that exposure to risks is held equal across all individuals, irrespective of the distinct qualities of the user themselves. This, from a theoretical standpoint is at odds with the general view of risky behavior; even that which occurs offline. For example, it is noted that the demographic factors of the user are related to the likelihood that they will be exposed to violent extremist material (even unsolicited) [29]. What this means then, and in line with an interactionist perspective of risky online behavior, is that factors associated with the individual, and indeed their internet tendencies, will play an important role in the nature of the risk that they are exposed to. This proposition makes intuitive sense, given that it is known that when it comes to the negative effects of harmful media, susceptibility stems from three discrete sources; dispositional, developmental, and social susceptibility. Thus in order to fully conceptualize the risks of being “online”, we must disaggregate both the nature of risks that individuals are exposed to, and the differential effect such harmful material may have on them [30,31,32]. This interactionist and disaggregated view of risk exposure online for young individuals is currently not fully explored but is critical in understanding how best to prevent harmful outcomes of being online.

### 1.2. Current Study

This study was conducted in a practice context where a group of schools decided to participate in an educational initiative designed to reduce hate and enhance tolerance towards diversity. Prior to the implementation of this educational initiative, a survey was conducted to understand students’ behaviors and attitudes, exposure to hate messages, and exposure to risky situations online. The analysis we present in this manuscript includes a subset of the survey questions focused on online behaviors and exposure to risky situations. This analysis was descriptive in its intent-we sought to describe predictors of adolescents’ exposure to online risks in the sampled population with no intention of deriving conclusions regarding the general population of 8th and 9th graders in the selected schools’ districts, the state of Utah, or the USA. The study was guided by the following research questions: What demographic characteristics are associated with exposure to online risks? How is the use of social media and private messaging apps associated with exposure to online risks? To what extent is adult supervision of adolescents’ online activities a protective factor towards exposure to online risks?

## 2. Materials and Methods

### 2.1. Study Design

We implemented a cross-sectional study design by the use of an online survey, in four schools in Utah (USA), gathering data from students in 8th and 9th grade in April 2018. The consent process contained two stages-one month before the start of data collection, parents were provided the study goals and opt-out forms. Students whose parents opted out were not invited to take the survey. We then obtained consent from the remaining students at the beginning of the survey. Data were collected using the online survey platform Qualtrics. The survey link was shared with the students by their teachers during class time–the survey took roughly twenty minutes to complete. We opted for a short survey to avoid disruption of learning activities as requested by the teachers. The study protocol and instruments were approved by the Harvard T.H. Chan School of Public Health Institutional Review Board (IRB16-1757; 24 January 2017) as well as by the ethical committee of the school district where the study was implemented. The Helsinki ethics protocols were followed throughout the course of this study [33]. A copy of the survey instrument can be found in the Appendix A.

### 2.2. Dependent Variables

To measure exposure to online risks, respondents were asked if they had ever experienced any of the following eight situations: (1) Bullying or harassment by friends or acquaintances, (2) Getting involved in unwanted conversations in a chat room, social networking site, or on email, (3) Coming across sexual images or content, (4) Coming across images of violence, (5) Someone trying to sell me drugs or alcohol, (6) Someone using my photos in an inappropriate way, (7) A stranger trying to meet with me, (8) Coming across hate groups trying to convince me of their views. Some of these situations can be classified as content risk (q3 and q4), some others as contact risk (q1, q2, q6, q7, and q8), and one as criminal risk (q7). This dependent variable was studied in two ways: (1) using each of the eight situations as an outcome, (2) using the number of situations a student encountered as the outcome. We also created a dichotomous variable where 0 = no situations encountered and 1 = any number of situations encountered, to further study and confirm the relationship found in the previous analysis.

### 2.3. Independent Variables

Independent variables included demographics (race, gender, and age), in addition to academic performance, use of social media and private messaging apps, and adult supervision. Academic performance was measured by frequency of grade type (A, A−/B+, B, B−/C+, and C or lower). Use of social media was measured by asking respondents which social media tool (i.e., YouTube, Snapchat, Instagram, etc.) they use and how frequently they use it, with answer options ranging from 1 (never) to 6 (all the time). A summative social media use score was created by dichotomizing the use of each social media platform used by the respondent into a dichotomous variable (1 = daily use or higher, 0 = less than daily use) and summing the attributed value across all fifteen platforms. Respondents were also asked if they used any private messaging apps, including Kik, Telegram, or WhatsApp, and if they used such apps more frequently than texting.

To further understand social media use and online behaviors, respondents were asked how many of their social media followers they knew in person, and how frequently they chatted with strangers while on social media. A similar question was asked about their friends’ behaviors, under the assumption that they would be less likely to misreport friends’ habits compared to their own. Additional questions focused on behaviors such as sharing personal information while on social media or chatting with strangers while playing online video games. Finally, using items adapted from existing research, respondents were asked about adult supervision, in particular, if their parents supervised their online activities, if anyone had ever talked to them about online safety, and if they had a trusted adult to ask for help in case they encountered an uncomfortable situation online [34].

### 2.4. Statistical Analyses

Simple and multiple logistic regression models were used to study how the independent variables were associated with the occurrence of each of the 8 situations identified as online risks. A Box-Tidwell procedure was used to confirm that aggregate social media use had a linear relationship to the log odds of each of the 8 dependent variables. Independent variables were included in the multiple models when a statistically significant association was found in the simple models (*p*-value < 0.05). Gender, race, and age were included regardless of their significance in the final models because of theoretical relevance. Hosmer-Lemeshow tests were used to assess the goodness of fit of the multiple logistic regression models [35]. For the second dependent variable related to the number of situations the respondents experienced, a negative binomial regression model was fitted to the counts of the risky online situations. Pearson’s chi-squared test of dispersion was used to assess the goodness of fit for the negative binomial model. We compared the fit of the negative binomial regression model to Poisson, zero-inflated Poisson, and zero-inflated negative binomial models using the Bayesian information criterion (BIC) to determine whether any of those distributional forms was more appropriate for this sample. For both logistic and negative binomial regressions, model misspecification was assessed using Pregibon’s goodness-of-link test [36]. For models where the goodness-of-fit test failed, we tried to modify the set of covariates in the model to achieve a better fit as feasible. Data analysis was performed using Stata Statistical Software Release 15.1 (StataCorp LLC, College Station, TX, USA).

## 3. Results

The response rate to the survey was 86% (767/897). This paper is based on the analysis of the data obtained from 733 students of which we had complete data. Below we provide the sample characteristics based on demographics and academic performance, and descriptive statistics for social media use, private messaging use, online behaviors, adult supervision of online activities, and risky online situations experienced by the respondents. We then present the results of the multiple logistic regression models for the dependent variables (each of the eight risky situations and exposure to at least one situation). Finally, we present the results of the negative binomial regression model for the count of different risky online scenarios.

### 3.1. Sample Characteristics and Descriptive Statistics

Detailed descriptive statistics for the sample can be found in Table 1. We gathered data from a convenience sample of 733 individuals, the majority of whom were male (51%). Major categories of race were: White (39%), Hispanic (21%), and mixed-race (10%). The sample included students aged 14 (41%), 15 (58%), and 16 (1%). Regarding academic performance, most students reported receiving grades ranging between A– and B+ (41%).

#### 3.1.1. Social Media Use and Online Behaviors

Detailed descriptive statistics for the sample can be found in Table 1. Most participants reported using more than one social media platform daily (85%), and 50% reported using three or more social media platforms daily. Only 14% of students had met all of their social media followers in person, the majority of them (62%) reported that they had chatted with strangers on social media, and 62% reported they believe their friends did so as well. Thirty-one percent of respondents indicated they had shared personal information, such as their school or town name when posting on social media. Eleven percent of students used private messaging apps more than texting, and use was distributed as follows: WhatsApp (9%), followed by Kik (5%), and Telegram (1%). The majority of participants played video games online (78%), and 51% reported that they chatted with someone they did not know while gaming. Regarding parents’ supervision, the majority (52%) of students reported that their parents did not install parental controls on their computers or other devices. The majority (69%) of students reported they had a trusted adult they could ask for help if they experienced an online situation that made them feel uncomfortable. Finally, 95% of respondents reported that someone had spoken to them about online safety.

#### 3.1.2. Risky Online Situations

Table 2 presents the detailed summary measures of each risky online event and the count of risky online events.

The most frequently reported situation was coming across images of violence (32%), followed by coming across sexual images or content (31%), and bullying by friends or acquaintances (29%). Twenty-six percent of respondents indicated they had been in an unwanted conversation in a chat room, social networking site, or on email. Seventeen percent of students reported experiencing a situation in which they encountered a stranger online who wanted to meet with them, and 14% of students reported coming across hate groups trying to convince them of their views. Fourteen percent of students indicated that someone had tried to sell them drugs or alcohol online. Only six percent of students reported someone inappropriately using their photos. Overall, the distribution of the counts of risky online situations was skewed to the right with subjects reporting a mean of 1.7 events (SD = 2) and a median of 1 (range = 0–8). Forty percent of the subjects did not report experiencing any of the risky online situations listed in the survey.

### 3.2. Logistic Regression Models

Table 3 presents the results for the multiple logistic regression models for each risky online event. Simple logistic regression models were performed for each of the dichotomous dependent variables. Box-Tidwell test results confirmed that our summative measure of overall social media use was linear to the log odds of each outcome (*p* > 0.05). Detailed results for the simple models can be found in the Appendix A. The overall LR chi-square test statistics for the eight multiple models exploring the association between the independent variables and each of the eight online situations were significant (χ^2^, *p* < 0.01). Hosmer-Lemeshow Goodness of Fit test results confirmed that all models were a good fit for the data. Pregibon’s link test did not indicate any model misspecification (ŷ *p* < 0.05, ŷ^2^ *p* > 0.05), except for the dependent variable “someone trying to sell me drugs or alcohol”. Below we provide a summary of key results for each situation and the counts of situations experienced.

#### 3.2.1. Online Content Risk

Two of the eight risky online situations we investigated referred to exposure to harmful contents (content risk): (1) coming across sexual images or content and (2) coming across images of violence. Gender was predictive of exposure to both of these risks. Compared to males, females had twice the odds of coming across sexual images or content (OR = 2, 95% CI 1.4–2.9) and 1.9 times the odds of coming across images of violence (OR = 1.9, 95% CI 1.3–2.7). Chatting with a stranger while online was also a predictor for both of these content risks. Respondents who reported this behavior had 70% increased odds of coming across sexual content (OR = 1.7, 95% CI 1.1–2.5) and images of violence (OR = 1.7, 95% CI 1.1–2.6). Race was the only other variable that predictive of exposure to sexual images; white students had 50% increased odds of coming across this content compared to non-white students (OR = 1.5, 95% CI 1–2.1).

Regarding the other content risk, age, academic performance, and social media use predicted coming across images of violence. Students aged 15 and older had 70% increased odds of coming across images of violence (OR = 1.7, 95% CI 1.2–2.4). Students who received grades of B or lower had 60% increased odds of coming across images of violence, compared to those who had grades of B+ or higher (OR = 1.6, 95% CI 1.1–2.3). Finally, the odds of coming across images of violence increased by 24% with each additional social media platform used by the respondent daily (OR = 1.24, 95% CI 1.08–1.42).

#### 3.2.2. Online Contact Risk

Five out of the eight online risky situations we investigated referred to interactions with someone online that lead to a risky outcome (contact risk situations): (1) bullying and harassment by friends or acquaintances, (2) getting involved in unwanted conversations, (3) someone using the respondent’s photos in an inappropriate way, (4) a stranger trying to meet with them, and (5) coming across hate groups trying to convince the respondent of their views. Chatting with a stranger while online was a predictor for all of the five contact risks. Respondents who reported this behavior had increased odds of exposure to the five contact risk situations. These increased odds ranged from 80% for having experienced unwanted conversations (OR = 1.8, 95% C.I. 1.2–2.9), and for coming across hate groups (OR = 1.8, 95% C.I. 1–3.2) to 350% for having had someone inappropriately use the respondent’s photos (OR = 4.5, 95% C.I. 1.5–13.3). Gender was associated with two of the five situations, with females having 70% increased odds of experiencing bullying or harassment while online (OR = 1.7, 95% 1.2–2.4) and 100% increased odds of engaging in unwanted conversations (OR = 2, 95% C.I 1.4–2.9). The number of social media platforms used by the respondent was associated with experiencing bullying or harassment in the online space, the odds of experiencing bullying or harassment increased by 14% with each additional social media platform used by the respondent daily (OR = 1.14, 95% CI 1.01–1.28). Finally, the use of the private messaging app Kik was associated with 210% increased odds of having someone inappropriately use the respondent’s photos (OR = 3.1, 95% CI 1.1–9.2), however, this association should be explored in a larger sample, as only 37 students in our sample used Kik.

#### 3.2.3. Criminal Risk

One of the eight situations we investigated referred to criminal risk: someone trying to sell the respondent drugs or alcohol. None of the independent variables resulted to be a significant predictor of this situation.

### 3.3. Negative Binomial Model for Number of Types of Risky Online Situations

Table 4 displays the results of the negative binomial model for the number of types of risky online situations. The final negative binomial regression model had a Pearson’s chi-square of 739.8 (df = 693, *p* = 0.11) indicating the model accounted for any overdispersion in the data well. Pregibon’s link test did not indicate any model misspecification (ŷ *p* < 0.01, ŷ^2^ *p* = 0.36). The model included the following factors: age, gender, race, social media use, chatting with strangers on social media, the respondent’s friends’ habit of chatting with strangers on social media, parents’ supervision of online activities, if anyone had ever talked to them about online safety and if they have a trusted adult to ask for help with risky online situations. The negative binomial model had a BIC of 2445.6 which was favorable when compared to the BIC from a Poisson (BIC = 2646.4), zero-inflated Poisson (BIC = 2487.5), and zero-inflated negative binomial (BIC = 2460.7) models with similar predictors.

In the negative binomial model, females had 1.5 times the incidence rate of experiencing different types of risky online situations compared to males (IRR = 1.5, 95% CI 1.3–1.9). Social media usage was also a significant predictor with each additional social media platform used daily associated with an 8% increase in the incidence rate of experiencing different types of risky online scenarios (IRR = 1.08, 95% CI 1.01–1.15). Students who chatted with strangers on social media had 1.6 times the incidence of experiencing different types of risky online scenarios than those who did not (IRR = 1.6, 95% CI 1.3–2.0). Lastly, the frequency in which the respondent’s friends chatted with strangers on social media was a significant predictor (χ^2^(df) = 14.6(3), *p* = 0.002). This was driven by significantly higher incidence rates of experiencing different types of risky online scenarios for students whose friends “sometimes” or “often” chatted with strangers on social media compared to students who did not know if their friends chatted with strangers with an IRR of 1.5 (95% CI 1.2–1.9) and 1.6 (95% CI 1.2–2.1), respectively.

Similar results were obtained in logistic regression models for our dichotomized variable for exposure to any risky online situation (see Table 3). Results from these analyses similarly indicated that gender (OR = 2.0, 95% CI 1.4–2.7), social media use (OR = 1.16, 95% CI 1.01–1.32), chatting with strangers online (OR = 2.3, 95% CI 1.6–3.4), and friends chatting with strangers online (χ^2^(df) = 11.8(3), *p* = 0.008) was significantly associated with exposure to risky events online.

## 4. Discussion

The goal of this study was to explore the individual-level characteristics associated with adolescents’ exposure to online risks, with a specific focus on demographic characteristics, various online behaviors, including use of social media and private messaging apps, and the role of adult supervision of adolescents’ online activities as a protective factor towards exposure to online risks. Findings from our study suggest how some children’s characteristics and behaviors are associated with the likelihood of encountering risky or uncomfortable situations when on the internet. Specifically, the type and duration of social media activity were strongly associated with exposure to some forms of online risk even if a temporal and causal association could not be investigated due to the cross-sectional study design. There was little evidence of parental supervision as a protective factor.

One particularly interesting result from our study is the existence of gender disparities in the exposure to online risks, with girls being more likely than boys to encounter specific risky situations in the online space. The role of gender is inevitably complex and multifaceted, but our data show that such disparities seem to be independent of other online behaviors such as the amount of time spent online or type of social media platform being used, suggesting that social factors may play a role including the potential for girls being the target of harassment, violence and sexual abuse. Our results should be consumed with the results from other studies in mind-specifically those that explore the social context that may give rise to these gender disparities in online risk. While some extant research has identified protective population-level factors against online risks, such as a country’s level of Internet diffusion, it is also vital to recognize studies that demonstrate online gender disparities in risk also occur in settings where offline exposure to risk and violence is gendered as well [37,38,39,40]. Future research should continue to investigate how societal factors may influence gender differences in online exposure to harmful material.

### 4.1. Implications for Policy Making

At the global level, we are increasingly focused on the harm caused by material accessed on the internet. Policymakers, academics, politicians, mental health, and medical professionals seek to understand the processes that govern the relationship between internet use and risk. Overall, the theoretical importance of this paper and its policy implications are related to its focus on the role of the individual, their behaviors, and characteristics, as a correlate to exposure to risky online events. While the majority of extant research has investigated the effect of the internet on the individual, such as the induced feeling of anonymity [27], researchers are increasingly exploring the role of the *user* on the likelihood that they engage in or are exposed to risky material online [30,41]. This research reinforces this interactionist perspective of online risk in that exposure to online risk is the result of an interaction between the nature of “being online” with the pre-existing characteristics, behaviors, and personality of the user.

Our findings also support the idea of differential risk in that while certain characteristics have a domain-general relationship with risk, others are associated with risk in a refined (or domain-specific) manner. For example, while being female is associated with the risks of being exposed to bullying, violent and sexual images, unwanted conversations, and at least one risky event, our data suggest that using the private messaging app Kik is associated with an individual’s photos being used inappropriately. However, we note that the use of Kik should be further investigated in larger studies due to the small number of individuals using this platform in our sample.

From a theoretical perspective, our results suggest the need to better differentiate the relationship between characteristics, and or social media use and the specific type of risk that someone may encounter. From a behaviorist standpoint, extant literature has shown that broad patterns of behavior reflect different fundamental incentives to human action [42,43]. This means that when it comes to online risky situations, including criminal activity, future research should aim to investigate the underlying incentives that guide both youth engagement with certain forms of online behavior and the certain risks that may stem from this. It is critical that forms of online intervention begin to think not just about domain-general aspects of prevention that can be equally beneficial to all (e.g., online safety, online disinhibition, parental rule or regulation), but also domain-specific forms of intervention that may need to be targeted towards certain types of risk that certain sub-groups of young internet users are more likely to be exposed to. These domain-specific forms of intervention must then include not only the specific online domain that individuals may be using (e.g., Kik) and the unique risks that this poses to these groups, but also the deeper psychological phenomena that are driving the online behavior.

The question of coping is important, as well as part of a wider shift from a technologically determinist discourse (of what the internet “does” to children) in favor of recognizing the importance of building children’s digital resilience. Encountering risk could in itself also represent an opportunity–to become more resilient, more digitally literate, and less vulnerable to online risks [44]. Sustained efforts are required from a variety of stakeholders, including families, schools, and technology industries to effectively safeguard children online in a dynamic technological and social media landscape. Thus, it is important to identify vulnerabilities that expose some children to higher risks so as to focus prevention efforts on those who will most benefit from them.

Future research should continue to investigate how societal factors may explain gender differences and other individual level-characteristics in online exposure to harmful material. We also recommend that future research attempt to understand how parents can reduce children’s vulnerabilities, whose role and supervision was not demonstrated as effective in our study.

### 4.2. Implications for Educational Initiatives

Educational initiatives should be tasked with developing programming for youth to address the underlying factors associated with online threats. These initiatives should be designed to develop the critical capability in children to recognize and mitigate the risks encountered by also learning how to self-regulate their behavior and seek help when needed. Children learning how to identify and interpret the impact and potential repercussions of various online behaviors can be influential in the overall safeguarding of youth online; among these potential risks, our study highlighted those posed by chatting with strangers, excessive use of social media platforms, and private messaging apps, and the development of both the technical and emotional competence to self-regulate online behaviors and deal with uncomfortable situations.

Initiatives that build knowledge and awareness about online risks are necessary, as is better awareness of existing disparities in exposure to risks. What this research can lend to the building and framing of educational internet safety initiatives is how to talk about and approach particular risks with students. Each risk online should be approached and discussed uniquely and based on individual risks, avoiding “one-size-fits-all” narratives that assume that all risks and the nature of such risks are the same for every internet user. Internet safety programs, such as those that are most prevalent in the education space like I-SAFE, Common Sense Media, and Netsmartz, could benefit from these findings in their efforts to avoid the use of “one-size-fits-all” approaches. In a multi-year study that evaluated internet safety education programs-such as those above-and their curriculum, researchers identified (among many other findings) that the most effective internet safety education programs need to have different strategies for different internet safety topics [45]. There is a great need to understand the risk factors and elements that might impact what risks and threats youth may be exposed to on different parts of the internet. Similarly, with many topics within internet safety being complex, the more explicit and clear educational programs can be when talking about risks and where youth might encounter them, the more beneficial it is to the educational goals.

Our findings show that the risks associated with certain online messaging apps are not all-encompassing, but rather narrowly associated with the inappropriate use of ones photos. Furthermore, chatting with strangers on social media has an array of potential associated risks. Understanding this connection and the association of certain platforms, actions, and risks can help inform education programming to give a more direct educational lesson to students to help with improving their self-regulated behaviors, necessary skills, and understanding of the potential risks around them.

Educational organizations and schools should consider our findings when building lessons and their content; learning examples and worksheets can better reflect certain risks and their association with certain behaviors. Additionally, our results should inform the way targeted programming for those who may be disproportionately exposed to risks online should be handled. Our data show that gender is associated with exposure to risky or uncomfortable situations, therefore calling on prevention programs to address these concerns with catered content. Simple changes to existing curricula might include the changing of the subject in particular examples to reflect a more accurate depiction of the research above, or more large-scale changes might be to offer further educational programming for those populations that are shown to be at higher risk of certain exposures.

### 4.3. Study Limitations

In considering the findings of this study, several limitations are acknowledged. Since our data were cross-sectional, we lack information on the temporal ordering of our variables, and we cannot assume causality. As a non-random convenience sample, it is important to acknowledge that these results are not necessarily generalizable outside of the sample, and that selection bias may have also influenced our reported results. Furthermore, as some of the exposures and outcomes could be perceived as negative, there is potential for social desirability bias to influence students’ responses. Our study sought to explore associations between risky online events and various behaviors online, demographic characteristics, and parental activities. It was not our intent to create an index to measure an underlying construct of online safety, and as such, no psychometric properties are reported. To our knowledge, there is no extant screening tool for exposure to online risk that has been psychometrically evaluated–future research should aim to address this gap.

## 5. Conclusions

Our results provide critical information to practitioners involved in the development of educational initiatives by suggesting the need to identify individual-level characteristics and behaviors that are associated with exposure to online risks in the population targeted by their programs, so to allow them to tailor their initiatives to meet the needs of those more likely to encounter online risks. The results presented in this manuscript also provide a platform for future longitudinal research to further investigate the magnitude of risk associated with these behaviors. Our results should be interpreted under the growing understanding that exposure to risky material online should be understood as a matter of if, not when, and as such, we recommend that future research should investigate how individuals engage with risky online content and the psychological processes which may influence exposure. With the continued increase of youth internet use, the need for reflective, impactful internet safety education is becoming more crucial. Schools are the main means by which a society can address digital safety and citizenship issues for all children within a structured learning environment. To achieve optimal success in this area, it requires teacher training and curriculum materials that are age and gender-appropriate as well as further research on the factors associated with online vulnerability.

## Figures and Tables

**Table 1 ijerph-18-05786-t001:** Students’ Online Behaviors.

Behavior	*N* (%)
Chat with strangers on social media	
I don’t have a social media account	57 (8)
Never	222 (30)
I did it a few times	215 (29)
Sometimes	154 (21)
Often	85 (12)
Share personal information on social media	
I don’t have a social media account	76 (10)
Never	433 (59)
Sometimes	191 (26)
Often	33 (5)
Talked to about online safety	
Never	35 (5)
Sometimes	280 (38)
Often	418 (57)
Trusted adult	
No	79 (11)
Yes	507 (69)
Not sure	121 (16)
It depends on the situation	26 (4)
Presence of parental controls on computer or other devices	
No	378 (52)
Yes	236 (32)
Not sure	119 (16)
Use Kik	
No	696 (95)
Yes	37 (5)
Use Telegram	
No	724 (99)
Yes	9 (1)
Use WhatsApp	
No	666 (91)
Yes	67 (9)
Connections know in person	
I don’t have a social media account	82 (11)
I am not sure how many	130 (18)
Some of them	149 (20)
Most of them	272 (37)
All of them	100 (14)
Friends chat with strangers on social media	
I don’t know	195 (27)
Never	84 (11)
Sometimes	319 (44)
Often	135 (18)
Use private messaging more than texting	
No	581 (79)
Yes	78 (11)
Not sure	74 (10)
Chat with strangers while gaming	
I do not play video games	97 (13)
Never	260 (36)
Sometimes	215 (29)
Often	161 (22)
Play video games	
Never	163 (22)
Often	289 (40)
Sometimes	281 (38)
Number of social media platforms used daily	
0	112 (15)
1	87 (12)
2	165 (23)
3	188 (26)
4	111 (15)
5	52 (7)
6 or more	18 (2)

**Table 2 ijerph-18-05786-t002:** Risky Online Situations.

Situation	*N* (%)
Coming across images of violence	236 (32)
Coming across sexual images or content	224 (31)
Bullying or harassment by friends or acquaintances	214 (29)
Getting involved in unwanted conversations in a chat room, social networking site or on email	187 (26)
A stranger trying to meet with me	125 (17)
Coming across hate groups trying to convince me of their views	102 (14)
Someone trying to sell me drugs or alcohol	101 (14)
Someone using my photos in an inappropriate way	42 (6)
Number of Situations Encountered	
0	291 (40)
1	147 (20)
2	87 (12)
3	86 (12)
4	45 (6)
5	34 (5)
6	17 (2)
7	8 (1)
8	18 (2)
Mean (SD) = 1.7 (2)	
Median (Range) = 1 (0–8)	

**Table 3 ijerph-18-05786-t003:** Multiple Logistic Regression Models.

Covariate	OS 1 Bullying	OS 2 Unwanted Conversations	OS 3 Sexual Images	OS 4 Violent Images	OS 5 Drugs & Alcohol	OS 6 Photos Used Inappropriately	OS 7 Stranger	OS 8 Hate Groups	At Least One Risky Event
	**Continuous and Dichotomous Predictors–OR (95% CI)**								
Age (15 and older vs. 14)	1.1 (0.8–1.6)	1.2 (0.8–1.8)	1.2 (0.8–1.7)	1.7 (1.2–2.4) **	1.4 (0.8–2.2)	0.8 (0.4–1.6)	1.5 (0.9–2.2)	1.2 (0.8–2)	1.3 (0.9–1.8)
Gender (Female vs. Male)	1.7 (1.2–2.4) **	2 (1.4–2.9) **	2 (1.4–2.9) **	1.9 (1.3–2.7) **	1.4 (0.9–2.3)	1.8 (0.9–3.9)	1.6 (1–2.7)	1.6 (1–2.5)	2 (1.4–2.7) **
Race (White vs. Non-White)	1.3 (0.9–1.9)	1.1 (0.8–1.7)	1.5 (1–2.1) *	1.1 (0.8–1.6)	1.2 (0.7–1.9)	0.9 (0.4–1.9)	0.7 (0.5–1.1)	0.8 (0.5–1.3)	1.2 (0.9–1.7)
Grades (B and lower vs. B+ and higher)	-	-	-	1.6 (1.1–2.3) *	-	-	-	-	-
Kik Use (Yes vs. No)	1.8 (0.9–3.9)	-	1.9 (0.9–4)	-	1.2 (0.5–3)	3.1 (1.1–9.2) *	1.6 (0.7–3.8)	2.1 (0.9–4.9)	1.4 (0.6–3.2)
WhatsApp Use (Yes vs. No)	-	-	-	-	-	-	1.2 (0.6–2.3)	-	1.2 (0.6–2.2)
Telegram Use (Yes vs. No)	-	-	-	-	-	-	-	-	-
Chatting with strangers on social media (Any Frequency of Communication vs. Never/I don’t have a social media account)	1.9 (1.3–2.8) **	1.8 (1.2–2.9) *	1.7 (1.1–2.5) *	1.7 (1.2–2.6) *	1.7 (1–3.1)	4.5 (1.5–13.3) *	2.1 (1.2–3.4) *	1.8 (1–3.2) *	2.3 (1.6–3.4) **
Sharing Personal Information on Social Media (Any Frequency of Sharing vs. Never/I don’t have a social media account)	-	1.2 (0.8–1.8)	-	-	-	1.7 (0.8–3.5)	1.3 (0.8–2)	-	1.3 (0.9–1.9)
Playing video games (Any Frequency of Use vs. Never)	-	-	-	-	-	-	0.9 (0.5–1.6)	-	-
Chatting with strangers while playing video games (Any Frequency of Communication vs. Never/I don’t play video games)	-	-	-	-	-	-	0.5 (0.3–0.8) *^,a^	-	-
Having someone talk to you about online safety (continuous)	-	-	-	1.2 (0.9–1.6)	-	-	-	-	1.2 (0.9–1.6)
Social Media Use (continuous)	1.14 (1.01–1.28) *	1.07 (0.93–1.24)	1.11 (0.97–1.27)	1.24 (1.08–1.42) **	1.13 (0.95–1.35)	1.11 (0.86–1.43)	1.01 (0.87–1.18)	1.02 (0.86–1.22)	1.16 (1.01–1.32) *
	**Categorical Predictors with 3+ Categories–Chi-Squared (df); *p*-Value**								
Use Private Messaging Apps More than Texting (categorical)	-	-	-	-	-	-	-	-	-
Amount of followers known in person (categorical)	-	3.52 (4); *p* = 0.474	3.14 (4); *p* = 0.534	7.24 (4); *p* = 0.124	5.57 (4); *p* = 0.216	-	-	5.28 (4); *p* = 0.26	1.08 (4); *p* = 0.897
Frequency that friends chat with strangers on social media (categorical)	-	7.5 (3); *p* = 0.056	10.4 (3); *p* = 0.015	9.45 (3); *p* = 0.024	6.75 (3); *p* = 0.077	-	6.83 (3); *p* = 0.078	10.7 (3); *p* = 0.014	11.8 (3); *p* = 0.008
Having a trusted adult to ask for help with unsafe online situations (categorical)	6.78 (3); *p* = 0.079	3.91 (3); *p* = 0.271	-	-	-	4.35 (3); *p* = 0.226	-	-	-
Presence of parental controls on your computer (categorical)	-	-	-	-	-	-	-	-	-
Frequency that friends chat with strangers on social media	**Pairwise Comparison of Marginal Linear Predictors (If Applicable)–OR (95% CI)**								
“Never” vs. “I don’t know”	-	-	1.8 (0.9–3.4)	1.1 (0.6–2.2)	-	-	-	1.1 (0.4–2.9)	0.9 (0.5–1.6)
“Sometimes” vs. “I don’t know”	-	-	2.1 (1.3–3.3) **	1.5 (1–2.3)	-	-	-	1.2 (0.6–2.2)	1.8 (1.2–2.8) **
“Often” vs. “I don’t know”	-	-	1.5 (0.9–2.7)	2.2 (1.3–3.7) **	-	-	-	2.6 (1.3–5) *	1.6 (1–2.7)
“Sometimes” vs. “Never”	-	-	1.2 (0.6–2.2)	1.3 (0.7–2.5)	-	-	-	1.1 (0.4–2.9)	2 (1.1–3.5) *
“Often” vs. “Never”	-	-	0.9 (0.4–1.8)	2 (1–4)	-	-	-	2.4 (0.9–6.6)	1.7 (0.9–3.3)
“Often” vs. “Sometimes”	-	-	0.7 (0.5–1.2)	1.5 (1–2.3)	-	-	-	2.2 (1.3–3.7) *	0.9 (0.5–1.4)

* *p <* 0.05, ** *p <* 0.01. OS 1, bullying or harassment by friends or acquaintances; OS 2, getting involved in unwanted conversations in a chat room, social networking site or on email; OS 3, coming across sexual images or content; OS 4, coming across images of violence; OS 5, someone trying to sell me drugs or alcohol; OS 6, someone using my photos in an inappropriate way; OS 7, a stranger trying to meet with me; OS 8, coming across hate groups trying to convince me of their views. ^a^ Two respondents differentially classified their video game use by stating they had never played video games, and yet they had chatted with strangers online. To ensure the validity of our analysis, we re-constructed the model for question 7-a stranger trying to meet with you, without these two respondents included, and found no change in significance to the reported results. Pairwise marginal linear predictors were only computed on categorical predictors with 3+ categories that had significant Chi-Squared tests. Blanks cells for continuous, dichotomous, and categorical predictors with 3+ categories indicate that this variable was not significant in the simple models and thus not included in the multiple model. Blank cells for pairwise comparison of marginal linear predictors indicate that this categorical predictor with 3+ categories did not have a significant Chi-Squared test value. 3.2.2. Online Contact Risk.

**Table 4 ijerph-18-05786-t004:** Negative Binomial Model for Counts of Types of Risky Online Situations.

**Covariate**	**IRR (95% CI)**
Age (15 and older vs. 14)	1.1 (1.0–1.4)
Gender (Female vs. Male)	1.5 (1.3–1.9) **
Race (White vs. Non-White)	1.1 (0.9–1.3)
Kik Use (Yes vs. No)	1.2 (0.8–1.8)
Chatting with strangers on social media (Any Frequency of Communication vs. Never/I don’t have a social media account)	1.6 (1.3–2.0) **
Social Media Use (continuous)	1.08 (1.01–1.15) *
**Covariate**	**Chi-Squared (df); *p*-Value**
Frequency that friends chat with strangers on social media (categorical)	14.6 (3); *p* = 0.002
**Covariate**	**Pairwise Comparison of Marginal Linear Predictors (If Applicable)–IRR (95% CI)**
Frequency that friends chat with strangers on social media	
“Never” vs. “I don’t know”	1.3 (0.9–1.8)
“Sometimes” vs. “I don’t know”	1.5 (1.2–1.9) **
“Often” vs. “I don’t know”	1.6 (1.2–2.1) **
“Sometimes” vs. “Never”	1.2 (0.8–1.6)
“Often” vs. “Never”	1.2 (0.9–1.8)
“Often” vs. “Sometimes”	1.1 (0.8–1.3)

* *p* < 0.05, ** *p* < 0.01.

## Data Availability

The data gathered during the course of the project are available upon request, due to research subjects’ privacy considerations, via the national public repository of data Criminal Justice Data (NACJD), hosted by the Inter-university Consortium for Political and Social Research (ICPSR; at the University of Michigan), and can be found at the following link: https://www.icpsr.umich.edu/web/NACJD/studies/37338.

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
