# Peer review of "Adolescents’ Exposure to Online Risks: Gender Disparities and Vulnerabilities Related to Online Behaviors"

_ijerph, 2021, doi:10.3390/ijerph18115786_

Round 1

Reviewer 1 Report

Thank you for considering my comments. The way you have changed the text is very satisfactory. I have no more comments.

Author Response

Thank you so much for your feedback, we believe that it has greatly improved the manuscript.

This manuscript is a resubmission of an earlier submission. The following is a list of the peer review reports and author responses from that submission.

Round 1

Reviewer 1 Report

It is my pleasure to read this interesting study. It provides insight into the online risky behavior of adolescents. I had some suggestions to improve the manuscript.  

Since more risky behavior had been well studies, I will suggest the author need to review the previous finding to have a good comparison between this study and previous study.  

Are the informed consent had been gathered from all adolescents? 

The psychometric data of the questionnaire should be provided to prove their validity.  

Are all adolescents from one school? The limit on the generation of the result should be discussed for the convenience data. 

The demographic data should be compared with that of the general population.  

I will suggest listing the result according to the rank of their frequency.  

Since this is a cross-sectional study, it could not have a conclusion about predictors or risk factors. The discussion has too much causal conclusion which was unable to be supported by the data.  

Author Response

Reviewer 1

It is my pleasure to read this interesting study. It provides insight into the online risky behavior of adolescents. I had some suggestions to improve the manuscript.

Since more risky behavior had been well studies, I will suggest the author need to review the previous finding to have a good comparison between this study and previous study.

Yes, more risky behaviors have been studied and we now review more widely the risks that have been studied in previous literature and what is currently known about the prevalence of youth exposure to risky situations online and concerns of such risks on the internet that youth have. You will find the updated literature review on lines 91-122.

Are the informed consent had been gathered from all adolescents?

Yes, we apologize for failing to include this process. We added sentences to clarify the student consent process on lines 145-155.

The psychometric data of the questionnaire should be provided to prove their validity.

Thank you for this wise comment - we agree with the reviewer that research should focus on validating measures to capture the construct of online safety and that this represents a gap in the literature. Our intent in this paper was not to create an index to measure an underlying construct of online safety, but rather to explore associations between unsafe online events and various behaviors online, demographic characteristics, and parental activities. We have added sentences to the limitations and conclusion to capture this – you will find these changes on lines 532-539.

Are all adolescents from one school? The limit on the generation of the result should be discussed for the convenience data.

In section 1.2 Current Study (lines 125-135) we now explain the context in which the study was implemented - 4 schools in Utah embracing an educational initiative. We also explain that the study was not meant to derive conclusions on the general population of 8th and 9th graders in Utah or in the USA.

The demographic data should be compared with that of the general population.

This study was not designed with the intent of deriving conclusions on the general population which would have required a very different sample. However, the study internal validity is not affected by the type of sample. We now explain that generalizability was not the intent of our work – line 129-135.

I will suggest listing the result according to the rank of their frequency.

We have re-ordered Table 2, and section 3.1.2. such that the results are listed from most frequent to least frequent.

Since this is a cross-sectional study, it could not have a conclusion about predictors or risk factors. The discussion has too much causal conclusion which was unable to be supported by the data.

The discussion has been revised accordingly by re-emphasizing the limits of the study design and avoiding the use of the terms “predictors” and “risk” with regard to the characteristics we identified. Any mention of “online risk” refers to the risky nature of an unsafe online event, not that the characteristics we identified in our study are “risky” or “risk factors”. We thank the reviewer for this very important feedback.

Reviewer 2 Report

Although the topic is of interest, for media scholars the main question when reading this manuscript unfortunately remains 'What is new, what is the additional value in relation to previous, similar studies on online risks?". The authors do not clearly indicate their contribution to the field of media consumption and media effects among adolescents in relation to previous research. This is also reflected in an introduction section that does not mention large scale and longitudinal research into this field such as the work of Livingstone et al (cfr EU Kids Online projects, Yskills project, the latter is still running). The literature review remains far too limited and does not offer a true overview of important contributions and the latest progress within the interdisciplinary field of media studies and (mental) health related issues/developmental research among adolescents. In general: there is a lack of an appropriate theoretical framework in this manuscript. 

It is strange to speak of a study that is 'seeking to explore predictors of adolescents' exposure to online risks' and 'exploratory research questions' whereas the existing literature and hypothesis driven research is so widely studied during the last 15 to 20 years at least within media and public health studies. Based on the literature, this survey research should be hypotheses driven. When adding new possible predictors, an open explorative research question approach would be appropriate, but this is not the case (characteristics and media variables that are not new). 

Did parents provide an active parental consent or only the passive version (see mentioning of opt-out). 

Why did the researchers choose an online survey that only took 20 minutes? Where is Appendix A (no part of the manuscript)? 

Why did the researchers go for a convenience sample? 

Are the parental mediation questions validated (source, used in previous research, why did'nt the researchers opt for an often used parental media  mediation scale? 

There is a mismatch between the numbers/figures in table 1 and the description underneath 3.1.1: the text says that Whatsapp was use among 67% of the students but the table says 9% and n= 67. Also Kik application numbers are differently mentioned in the table then the text (37% vs 5%). The majority plays games online: 78% in text, 87% in table. Please recheck all numbers because this gives a very sloppy impression. 

3.1.2 'the majority of the subjects (40%) did not report...'. 40% is no majority.It can be the modus or the largest category though. 

3.2.2 contact risk: the use of private messaging with Kik is associated by increased odds of photo abuse. But the group of adolescents using Kik was small (5%) so is the n here not too small to generalize this conclusion? 

3.2.3 commercial risk: someone trying to sell drugs or alcohol: isn't there a better description for this type of risk? Is health related and goes against the law (criminal aspect). Commercial risk might be a too soft word choice (more related to advertising or risking ordering and paying products that will never be delivered etc). 

Discussion and conclusion: the authors talk about "differential risk". Therefore it would be good to widen the theoretical part in the introduction by including the DSMM model of Valkenburg and Peter (2013 and later publications) (Differential Susceptibility to Media effects Model). 

The discussion and conclusion suggests media literacy initiatives without giving examples of already existing initiatives. What do the authors advice on top of what is already available and what should be ameliorated based on their study? The discussion part remains far to vague on these points. 

The limitations of the current study should be described more clearly and widely pointing at future solutions. In the method part, the authors should explain why, although they knew about the shortcomings of cross sectional data collection and a convenience sample in advance, they opted for this approach. 

Author Response

Although the topic is of interest, for media scholars the main question when reading this manuscript unfortunately remains 'What is new, what is the additional value in relation to previous, similar studies on online risks?". The authors do not clearly indicate their contribution to the field of media consumption and media effects among adolescents in relation to previous research. This is also reflected in an introduction section that does not mention large scale and longitudinal research into this field such as the work of Livingstone et al (cfr EU Kids Online projects, Yskills project, the latter is still running). The literature review remains far too limited and does not offer a true overview of important contributions and the latest progress within the interdisciplinary field of media studies and (mental) health related issues/developmental research among adolescents. In general: there is a lack of an appropriate theoretical framework in this manuscript.

We agree with the reviewer that this introduction and theoretical framework was limited on the reflection to the existing literature in the field. We have addressed this by adding in some of Livingstone et al.’s work on youth’s perception of risk, Livingstone and Helsper’s work on parental mediation, and some of the multiyear analyses that have been done on internet and digital safety in EU adolescents. Lastly, we addressed the need for explanation in how this study furthers the field’s understanding of youth online behavior and risk - and specifically what it means for resulting actions in aiming to prevent future harms. You will find the updated literature review on lines 91-122.

It is strange to speak of a study that is 'seeking to explore predictors of adolescents' exposure to online risks' and 'exploratory research questions' whereas the existing literature and hypothesis driven research is so widely studied during the last 15 to 20 years at least within media and public health studies. Based on the literature, this survey research should be hypotheses driven. When adding new possible predictors, an open explorative research question approach would be appropriate, but this is not the case (characteristics and media variables that are not new).

We agree with the reviewer and have better explained how this study fits with the existing

literature in the introduction (lines 91-122) and have discussed the implications of the findings in the discussion (lines 443 – 455). We have also changed the language from exploratory to descriptive as this was the intent of the study – line 131-132.

Did parents provide an active parental consent or only the passive version (see mentioning of opt-out).

Yes, we apologize for failing to clarify this process. We added sentences to clarify the student consent process on lines 145-155.

Why did the researchers choose an online survey that only took 20 minutes? Where is Appendix A (no part of the manuscript)?

We now explain in section 1.2 (lines 125-135) and 2.1 (lines 157-158) the context in which this survey was conducted. It was a practice context in which there was a need to limit disruption of learning. The “Appendix” was really supplementary material – we are not sure why you were not provided with it, but hopefully you can access it with this new submission!

Why did the researchers go for a convenience sample?

In section 1.2 (lines 125-135) we explain that this study was conducted in the context of an

educational activity as such it had constraints in its implementation due to the need to avoid

disruption of learning in schools.

Are the parental mediation questions validated (source, used in previous research, why didn’t the researchers opt for an often used parental media mediation scale?

We agree with the reviewers that when possible, research should strive to use validated

measures. Unfortunately, because of the strict time constraints we were given by the schools’ administrations and teachers - 20 minutes, we could not include a measure as long as the parental media mediation scale. The items we used were adapted from previous research, however – we have now added the reference – line 200.

There is a mismatch between the numbers/figures in table 1 and the description underneath 3.1.1: the text says that Whatsapp was use among 67% of the students but the table says 9% and n= 67. Also, Kik application numbers are differently mentioned in the table then the text (37% vs 5%). The majority plays games online: 78% in text, 87% in table. Please recheck all numbers because this gives a very sloppy impression.

We apologize for this sloppy error- we have fixed the discrepancies between the table and the results for private messaging apps. The reported results for video game usage are correct - after reviewing the data - all but two of the students who selected “Never” for video game usage either selected “Never” or “I do not play video games” in the following question about chatting with strangers while playing video games. As the reviewers will note in tables 2 and 3, in our statistical analyses, we recoded the variable for chatting with strangers such that the categories “never” and “I do not play video games'' were combined. To ensure the internal validity of the analysis, we re-constructed the model for question 7 - a stranger trying to meet with you - without these two respondents included and found no change in significance to the reported results. We have also added a footnote to table 2 explaining the above – line 293. We thank the reviewers for their careful attention to detail on this matter.

3.1.2 'the majority of the subjects (40%) did not report...'. 40% is no majority. It can be the modus or the largest category though.

Thank you for your attention to this detail. We have changed the wording in the results where the reported percentage is below 50.

3.2.2 contact risk: the use of private messaging with Kik is associated by increased odds of photo abuse. But the group of adolescents using Kik was small (5%) so is the n here not too small to generalize this conclusion?

This limitation was addressed in the results (337-338) and discussion (436-438).

3.2.3 commercial risk: someone trying to sell drugs or alcohol: isn't there a better

description for this type of risk? Is health related and goes against the law (criminal

aspect). Commercial risk might be a too soft word choice (more related to advertising or risking ordering and paying products that will never be delivered etc).

We agree with the reviewer and thank them for the thoughtful feedback- we renamed it “criminal risk’

Discussion and conclusion: the authors talk about "differential risk". Therefore, it would be good to widen the theoretical part in the introduction by including the DSMM model of Valkenburg and Peter (2013 and later publications) (Differential Susceptibility to Media effects Model).

The authors thank the reviewer for highlighting this theory and absolutely agree with its relevance. We have incorporated a note referencing DSMM in the introduction (line 111-119) and the idea that risk is not equal among all, but differential based on preexisting dispositional, developmental, and social susceptibility factors. In addition to this, in the discussion we have reinforced that the existence of differential exposure (both in terms of susceptibility and specific risk manifestations) has important implications for the nature of interventions that are required especially in groups who exhibit specific dispositions that are associated with exposure to unsafe online events – see lines 443-455 and 488-499.

The discussion and conclusion suggests media literacy initiatives without giving examples of already existing initiatives. What do the authors advice on top of what is already available and what should be ameliorated based on their study? The discussion part remains far too vague on these points.

We agree with the reviewer on this and have added in the discussion two points: first is a reflection of who the leaders in the field of internet safety education programming are; second is how this research can be incorporated to some of the lessons already learned by these programs through previous evaluations to improve their product and (subsequently) keep youth safer online – see lines 488-499.

The limitations of the current study should be described more clearly and widely pointing at future solutions. In the method part, the authors should explain why, although they knew about the shortcomings of cross-sectional data collection and a convenience sample in advance, they opted for this approach.

In section 1.2 (lines 125-135) that this study was conducted in the context of an educational activity, practice-based context. The primary intent of that educational activity was not research.

Reviewer 3 Report

Dear Authors,

It was a great pleasure to read your article. The topic is very actual and extremely important. And – what the most important – study design and research realization are very satisfactory.

I have only two tips to improve the text.

One is connected with research questions. I suggest avoiding settlement questions, that is why I propose to change:

Is adult supervision of adolescents’ online activities a protective factor towards exposure to online risks?

to:

To what extent adult supervision of adolescents’ online activities is it a protective factor towards exposure to online risks?

Second:

Gender is highlighted in the subject of the text. It should also be more exposed in the abstract. In the Introduction and Discussion part should be more references to literature on gender differences on online risk, and studies in which were diagnosed differences in experiencing violence due to gender (not only online), e.g .:

Susanne E. Baumgartner, Sindy R. Sumter, Jochen Peter and Patti M. Valkenburg. Identifying Teens at Risk: Developmental Pathways of Online and Offline Sexual Risk Behavior. Pediatrics. December 2012, 130 (6) e1489-e1496; DOI: https://doi.org/10.1542/peds.2012-0842

Notten N, Nikken P. Boys and girls taking risks online: A gendered perspective on social context and adolescents’ risky online behavior. New Media & Society. 2016;18(6):966-988. doi:10.1177/1461444814552379

Ayala A, Vives-Cases C, Davó-Blanes C, et.all. Sexism and its associated factors among adolescents in Europe: Lights4Violence baseline results. Aggress Behav. 2021 Feb 21. doi: 10.1002/ab.21957.

Debowska A, Boduszek D, Jones AD, Willmott D, Sherretts N. Gender-Based Violence-Supportive Cognitions in Adolescent Girls and Boys: The Function of Violence Exposure and Victimization. Journal of Interpersonal Violence. 2021;36(3-4):1233-1255. doi:10.1177/0886260517741628

Thank you for considering my comments

Author Response

Reviewer 3

Dear Authors,

It was a great pleasure to read your article. The topic is very actual and extremely important. And – what the most important – study design and research realization are very satisfactory. I have only two tips to improve the text.

One is connected with research questions. I suggest avoiding settlement questions, that is why I propose to change: Is adult supervision of adolescents’ online activities a protective factor towards exposure to online risks? to: To what extent adult supervision of adolescents’ online activities is it a protective factor towards exposure to online risks?

Thank you for this helpful suggestion - we have made the change.

Second:

Gender is highlighted in the subject of the text. It should also be more exposed in the abstract. In the Introduction and Discussion part should be more references to literature on gender differences on online risk, and studies in which were diagnosed differences in experiencing violence due to gender (not only online), e.g .:

Susanne E. Baumgartner, Sindy R. Sumter, Jochen Peter and Patti M. Valkenburg. Identifying Teens at Risk: Developmental Pathways of Online and Offline Sexual Risk Behavior. Pediatrics. December 2012, 130 (6) e1489-e1496; DOI: https://doi.org/10.1542/peds.2012-0842

Notten N, Nikken P. Boys and girls taking risks online: A gendered perspective on social context and adolescents’ risky online behavior. New Media & Society. 2016;18(6):966-988. doi:10.1177/1461444814552379

Ayala A, Vives-Cases C, Davó-Blanes C, et.all. Sexism and its associated factors among adolescents in Europe: Lights4Violence baseline results. Aggress Behav. 2021 Feb 21. doi: 10.1002/ab.21957.

Debowska A, Boduszek D, Jones AD, Willmott D, Sherretts N. Gender-Based Violence-Supportive Cognitions in Adolescent Girls and Boys: The Function of Violence Exposure and Victimization. Journal of Interpersonal Violence. 2021;36(3-4):1233-1255. doi:10.1177/0886260517741628

Thank you so much for highlighting this opportunity to call to attention the greater social context at hand. We have added the below as references to the conclusion and attempt to highlight the need to explore the societal-level offline factors that give rise to or reduce gender disparities in online exposure to harm. You will find the changes on lines 401-408.